# Feature identification in time-indexed model output

**Justin Shaw** **\*, Marek Stastna**

Department of Applied Mathematics, University of Waterloo, Waterloo, Ontario, Canada

\* justin_shaw@outlook.com

## Abstract

We present a method for identifying features (time periods of interest) in data sets consisting of time-indexed model output. The method is used as a diagnostic to quickly focus the attention on a subset of the data before further analysis methods are applied. Mathematically, the infinity norm errors of empirical orthogonal function (EOF) reconstructions are calculated for each time output. The result is an EOF reconstruction error map which clearly identifies features as changes in the error structure over time. The ubiquity of EOF-type methods in a wide range of disciplines reduces barriers to comprehension and implementation of the method. We apply the error map method to three different Computational Fluid Dynamics (CFD) data sets as examples: the development of a spontaneous instability in a large amplitude internal solitary wave, an internal wave interacting with a density profile change, and the collision of two waves of different vertical mode. In all cases the EOF error map method identifies relevant features which are worthy of further study.

**Data Availability Statement:** We have uploaded all MATLAB generated Figures (2 through 7), and all MATLAB codes for these Figures. Here is the DOI for the GitHub repo for the University of Waterloo: (https://git.uwaterloo.ca/j9shaw/PLOS-one-2019.

## 1 Introduction

We present a data-centric diagnostic for identifying time subsets of model output which are worthy of further study. To minimize the cost of uptake and maximize the clarity of the presentation we have built this diagnostic on Empirical Orthogonal Functions (EOFs), which are used in an enormous variety of contexts (e.g. [1], [2], [3], [4], [5], etc.) and have implementations in every commonly used software toolbox (e.g. Matlab, R, Scipy). The method presented here can be applied to any data set for which an EOF analysis would be appropriate. However, we will focus on the application to CFD data sets. The method is data driven, using a novel construction: a map of the EOF reconstruction errors as a function of time and the number of modes in the reconstruction. The interpretation of this EOF error map yields the identification of interesting times in each field in the data set for the cost of one Singular Value Decomposition (SVD) and one norm calculation per time output and choice of reconstruction.

The mathematical ideas behind EOFs have a long history, originating with [6], and go by many names, including Principal Component Analysis (PCA), Singular Value Decomposition (SVD), and Principal Orthogonal Decomposition (POD), depending on the community. These methods produce an orthogonal basis for the state space of a data set, where the basis vectors (EOFs) are rank-ordered by the amount of variance of the data they capture, as

**Funding:** The author(s) received no specific funding for this work.

**Competing interests:** The authors have declared that no competing interests exist.

recorded in the eigenvalue for each basis vector. In particular, when the data has units of velocity, the variance has units of energy, so the basis is rank ordered by energy captured. Following the common parlance, we will use "energy" and "variance" interchangeably. Since the use of all basis vectors fully reconstructs the data, and the basis is rank-ordered by energy content, this representation can then be truncated to provide a reduced order reconstruction of the data. This reconstruction captures the most energy contained in the original data set per basis vector added, on average [7]. Efficient reconstructions of data are often the goal in statistical analysis, where EOF methods are referred to as PCA. For a review from this perspective see [8].

EOF methods are common in the atmospheric science, oceanography, and climate science communities where there has been an attempt to relate individual EOFs either to physical processes or to normal modes of the system being sampled. Such efforts have had some success, for example in the study of the El Niño Southern Oscillation [9], North Atlantic Oscillation [10], and the Arctic Oscillation [11]. The focus on the first, or "leading", EOF can be viewed as the study of a an EOF reconstruction (heavily) truncated to include only the first mode. As mentioned, some large scale motions have been captured this way, and correspondences have been drawn between physical processes and the leading EOF. However EOFs form an orthogonal set, and thus adding subsequent EOFs to the reconstruction, while simultaneously expecting those additional modes to correspond to physical processes, is to assume that the physical processes or normal modes in question are orthogonal. This is not true in general. Instead, a kind of contamination occurs: [12] applied an EOF analysis to a constructed flow with multiple dominant structures. They found that EOFs roughly corresponding to specific fluid structures were contaminated by components of other structures (their Figures 3 and 6). Several modifications to EOF methods have been developed to produce modes which may have a more direct physical interpretation, but these methods often require a choice to be made, and it is not often clear which choice is correct. We refer the reader to the review by [5] of EOFs and their extensions for a history of these difficulties. In the error map method we simply use the standard EOF, as it is the most widely used. Moreover, we focus on the reconstruction perspective in order to build the EOF error map. This avoids the difficulties of focusing on individual EOFs outlined above. In addition, the construction of the error map includes errors from every truncated reconstruction, so there is no need to consider the problem of choosing a particular EOF to focus on. Because it avoids focusing on either individual EOFs, or individual EOF reconstructions, the EOF error map method is different from every previous EOF-based method.

There are, of course, a wide variety of existing data analysis methods for CFD data sets which are not EOF-based, but none of them serve the same function as the EOF error map method presented here. There are local, Eulerian (i.e., measurements at fixed locations) methods to identify vortices based on the decomposition or invariants of the velocity-gradient tensor: the $Q$-, $\Delta$-, and $\lambda_2$-criterions for example [13]. There are Lagrangian methods (i.e., based on moving particles) to identify coherent structures (e.g. transport barriers), such as those based on Finite Time Lyapunov Exponents [14], [15], or graph theoretic methods [16], [17], [18]. For a comparison of multiple Lagrangian methods applied to the same benchmark see [19]. There are a host of methods based on the spectral properties of the Koopman operator [20], and its finite dimensional approximation the Dynamic Mode Decomposition [21], which allow identification of structures in fluid flows based on the frequency of the structure's motion, such as the flapping frequency of a jet [22]. There are many reduced order methods besides EOF, including the related POD and Galerkin projection [23], [7]. For a review see [24]. In fact, there are many more analysis methods available which can be used to study CFD data sets. All of them make an *a priori* judgement on the field of interest (e.g. gradient of the velocity field, inter-particle separation, etc) and proceed with an analysis on that particular

field in the data set. In contrast, the purpose of the EOF error map method is to identify interesting time periods within every field in the model output without an assumption on which variable is the most important. These features, in each field, then become targets for further study using any method appropriate, including those just mentioned.

Put another way, the EOF error map method is a diagnostic tool which is applied earlier in the analysis pipeline than the standard methods just discussed. As such it is not a competitor with those methods, but a way to facilitate their intelligent application. This is particularly relevant to large, coupled models in fields such environmental fluid mechanics involving biogeochemistry and climate modeling for which the CFD component is only a small portion of the model. Even sophisticated mathematical tools based exclusively on the fluid mechanics may miss an important event in one of the other components of the model (e.g. an algal bloom in the coupled model of a bay). Thus for large coupled models, we envision our method being applied as part of the model execution, so that every field in the model output would be accompanied by identified features. Only the subsequent analysis would be discipline specific.

Error maps also carry a very low overhead. They are constructed directly from model output immediately after the completion of a numerical experiment and the only extra computational burden is the SVD and error map construction: there is no need to take derivatives of fields, it is not necessary to have particle data, there is no necessity to tune parameters in a graph theoretic clustering algorithm, etc. Error maps are used as a diagnostic to quickly identify features which should be investigated further, by whatever method is deemed useful for the particular application. This allows error maps to inform decisions on where higher overhead methods should be applied. In summary, the EOF error map is a low overhead method applied directly to model output as a way of focusing the application of other methods.

The remainder of the paper is organized as follows. Section 2 gives a brief background on EOFs (2.1) and discusses truncated EOF reconstructions (2.2) before introducing EOF error maps (2.3). Section 3 applies the method to the three data sets: the development of a spontaneous instability (3.1), an internal solitary-like wave encountering a sharp change in the background density profile (3.2), and the collision of two waves (3.3). The results show that the EOF error map method is able to identify time periods of interest in CFD data sets. Section 4 is a discussion of the main conclusions to be drawn from this work and of possible extensions of the EOF error map method. Section 5 gives a brief summary and concludes the work.

## 2 Methods

### 2.1 Empirical orthogonal functions

We briefly review EOFs in order to set notation, and point out those facts that will be needed when introducing the error map method. Suppose the data set has $M$ grid points and $N$ time outputs at times $t_j$, $j = 1, \ldots, N$. This is a sequence of snapshots $\{\mathbf{x}(t_1), \mathbf{x}(t_2), \ldots, \mathbf{x}(t_N)\}$ where each $\mathbf{x}(t_j) \in \mathbb{R}^M$. Centre by the time mean, and make the resulting snapshots columns of a single matrix $\mathbf{X}$. Then the $j$th column of $\mathbf{X}$ is

$$\mathbf{X}_j = \mathbf{x}(t_j) - \langle \mathbf{x} \rangle \tag{1}$$

where the angle brackets indicates the time mean. The matrix $\mathbf{X}$ is

$$(\mathbf{X})_{ij} \quad = x_i(t_j) - \langle x_i \rangle \tag{2}$$

where $i$ indexes the grid points, $j$ indexes the time outputs, and $\langle \mathbf{x} \rangle_i = \langle x_i \rangle$. Then $\mathbf{X}$ is an $M$ by $N$ matrix whose entries are time mean-centred time series of measurements at the grid points. The standard derivation of EOF is often motivated by diagonalizing the covariance matrix of

$\frac{1}{N-1}\mathbf{XX^T}$ to obtain the EOF eigenmodes and eigenvalues $\lambda_k$. However we will instead present the SVD derivation, as the SVD method is generally more robust than the covariance matrix diagonalization method [25].

When $M \geq N$, as is common in CFD data, applying the SVD to $\mathbf{X}$ we obtain [26]

$$\mathbf{X} = \mathbf{U}\begin{bmatrix} \boldsymbol{\Sigma} \\ \mathbf{0} \end{bmatrix}\mathbf{V}^T \tag{3}$$

Where $\mathbf{U}_{M\times M}$ and $\mathbf{V}_{N\times N}$ are orthogonal matrices and $\boldsymbol{\Sigma}_{N\times N} = \text{diag}(\sigma_1, \ldots, \sigma_N)$. The columns of $\mathbf{U}$, $\{\mathbf{u}_1, \ldots, \mathbf{u}_N\} \subset \mathbb{R}^M$, are the orthonormal spatial EOF basis vectors (modes), where the $i$th entry $u_{ik}$ in the column vector $\mathbf{u}_k$ corresponds to the $i$th grid point of mode $k$. This basis corresponds to the singular values from $\boldsymbol{\Sigma}$ with

$$\sigma_1 \geq \cdots \geq \sigma_N \geq 0. \tag{4}$$

Carrying out the multiplication in Eq 3, we obtain [26]

$$\mathbf{X} = \sum_{k=1}^{r}\sigma_k\mathbf{u}_k\mathbf{v}_k^T$$

$$\mathbf{x}(t_j) = \sum_{k=1}^{r}\sigma_k v_{jk}\mathbf{u}_k + \langle\mathbf{x}\rangle$$

where $r = \text{rank}(\mathbf{X})$, and the second equation is the columnwise version of the first with the time output indexed by $j$. By multiplying both sides of Eq 3 by $\mathbf{U}^T$ we find that

$$\mathbf{u}_k \cdot (\mathbf{x}(t_j) - \langle\mathbf{x}\rangle) = \sigma_k v_{jk}, \tag{5}$$

so that the projection of the centred data onto the EOF basis yields time-dependent coefficients defined as

$$a_k(t_j) = \sigma_k v_{jk}. \tag{6}$$

Therefore the columns of $\mathbf{V}$, $\{\mathbf{v}_1, \ldots, \mathbf{v}_N\} \subset \mathbb{R}^N$, are the unscaled coefficients corresponding to each mode. The $j$th entry $v_{jk}$ in the column of $\mathbf{v_k}$ corresponds to the coefficient at time $j$ for mode $k$. The rank ordering of the singular values (Eq 4) becomes a rank ordering of the scaling of the $a_k$. The data can then be written as

$$\mathbf{x}(t_j) = \sum_{k=1}^{r}a_k(t_j)\mathbf{u}_k + \langle\mathbf{x}\rangle \tag{7}$$

Note that there are methods of producing EOFs which are dependent on time as well as space (see section 3.2 of [7]). The SVD method produces spatial EOFs and time dependent coefficients, which makes the interpretation of the error maps presented in section 2.3 and 3 completely straightforward. As mentioned this derivation was for the case $M \geq N$. In general the number of singular values is $\min\{M, N\}$, which is $N$ in the cases presented here. There is an analogous decomposition for $M < N$.

The submatrix of zeros in Eq 3 as well as the rank limited sum in Eq 7 both make it clear that at most the first $N$ modes $\mathbf{u}_1, \ldots, \mathbf{u}_N$ are needed. This leads to the reduced SVD [27], where $\mathbf{U}$ consists of only these columns and there is no submatrix of zeros with $\boldsymbol{\Sigma}$. We obtained this decomposition using MATLAB's built in `svds` command with $N$ modes recovered to avoid the memory constraints of `svd` (see the accompanying code and the MATLAB documentation for details). As mentioned, the SVD is a more stable algorithm for calculating

the modes and eigenvalues of the covariance matrix of **X**. The connection between the two is that the columns of **U** (up to sign) are the modes [7], and the nonzero eigenvalues and singular values satisfy $\lambda_k = \sigma_k^2$ [27]. To reduce notational clutter we have left out the scaling factor on **X** of $\sqrt{N-1}^{-1}$ in this presentation, but in practice it is included for equality with the covariance matrix. See section 15.4 of [25] for details. See [26] for more details on the SVD, and section 3.4.2 of [7] for more on the connection between SVD and the EOF.

## 2.2 Truncated EOF reconstructions

Eq 7 makes clear that the data can be thought of as a time mean vector signal with layers of corrections provided by the EOFs. This representation recovers the data completely, so that the error in the representation of the data set is at or near machine precision. However, the rank ordering of the singular values (Eq 4) implies that each successive mode added to the sum contributes less variance over time than the previous mode. To make this concrete, project the data at every time onto mode $k$ and sum:

$$
\begin{aligned}
&\sum_{j=1}^{N} \left| (\mathbf{x}(t_j) - \langle \mathbf{x} \rangle) \cdot \mathbf{u}_k) \mathbf{u}_k \right|^2 \\
&= \sum_{j=1}^{N} \left| (\mathbf{x}(t_j) - \langle \mathbf{x} \rangle) \cdot \mathbf{u}_k) \right|^2 |\mathbf{u}_k|^2 \\
&= \sum_{j=1}^{N} |a_k(t_j)|^2 \\
&= \sum_{j=1}^{N} , |\sigma_k v_{jk}|^2 \\
&= \sigma_k^2 \left( \sum_{j=1}^{N} |v_{jk}|^2 \right) \\
&= \sigma_k^2
\end{aligned}
\tag{8}
$$

where we've used the fact that **U** and **V** are orthogonal, along with Eqs 5 and 6. We see that the sum over time of the contributions of $\mathbf{u}_k$ is exactly the variance $\lambda_k = \sigma_k^2$. Note that this equation shows that the contribution $\lambda_k$ from $\mathbf{u}_k$ may be large either because of moderate contributions over most of the simulation, or large contributions over a short time, or some combination. The EOFs have been rank ordered by their total contribution to the reconstruction summed over time, but not by their contribution at any given time $t_j$. This time information has been summed out. This is related to the rank ordering of the singular values (Eq 4) providing a rank ordering in the scaling, but not a rank ordering of the values $a_1(t_j), \ldots, a_r(t_j)$ at any specific time $t_j$.

If $\sigma_i$ are small for some $i > D$ we can write

$$
\mathbf{x}(t_j) \approx \sum_{k=1}^{D} a_k(t_j) \mathbf{u}_k + \langle \mathbf{x} \rangle
\tag{9}
$$

so that truncated EOF reconstructions can be thought of as an energy or variance filter, because the $D$ EOF reconstruction captures $E = \sum_{k=1}^{D} \lambda_k$ of the energy. We consider only

these rank-ordered reconstructions of all modes up to and including *D*, for a total of *N* reconstruction for a given data set.

### 2.3 EOF error maps

With the background material clearly stated, we present the following novel construction. We are interested in finding features within model output fields which are worthy of further study. We will employ the SVD reconstructions just outlined to do so. As discussed in the introduction, individual EOFs do not generally relate to individual physical processes. However, every process contributes some amount to the total variance of the model output.

Consider the following thought experiment: rank order the (unknown) processes in the dataset by variance contributed. Just as Eq 8 shows that the contribution of an EOF to the reconstruction may be large either as a result of moderate contributions over a long duration or large contributions over shorter durations, so too the rank ordering of processes is the result of some combination of the size and duration of each process. We expect large variance processes to include those with large scales and long duration. We expect small variance processes to include those with short scales and short duration. In between are medium variance processes with large scales and short duration, small scales and long duration, or medium scales and duration. See Fig 1 for examples.

We wish to identify time periods of interest. This means short or medium duration, and for the phenomenon to be of interest, probably medium to large scale. This means we are looking for medium variance processes in the data set. However, as discussed, the EOF does not generally find processes individually. Instead, the contamination phenomenon described in [12] implies that as *D* increases the approximations of multiple processes are simultaneously

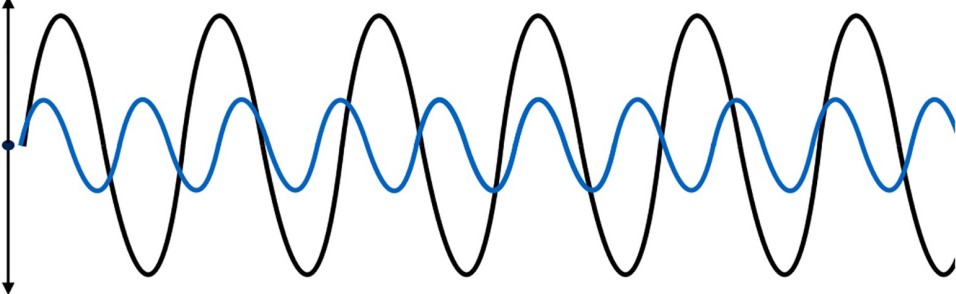

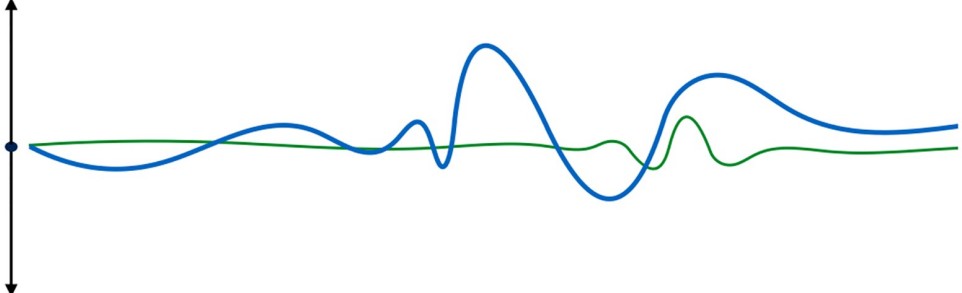

**Fig 1. Large, medium, and small variance processes.** Examples of large, medium, and small variance processes over time. The upper plot shows a large variance process which has a large scale and long duration, along with a medium variance process with less variance, but equal duration. The bottom plot shows a small variance process with a small scale and short duration, along with a medium variance process with larger scale and duration.

improved, and the higher the variance of a process, the greater its priority. Every mode added increases the variance represented rather than adding a process, but as variance represented increases more processes are approximated well. By convergence, some $D$ approximates all processes of interest. At the extreme end, if everything is of interest, $D = N$. Moreover the speed of convergence, as indicated in the scree (a plot of the eigenvalues), shows that higher modes essentially represent "noise" (here the quotations are included to indicate that we do not mean noise in the sense of stochastic processes). This means that some low choice of $D$ will tend to capture the large scales (as in the "elbow test", see [8]), while different choices of $D$ near $N$ are basically the same because the last modes in the decomposition have very small coefficients. Intermediate choices of $D$ will include those that poorly approximate a variety of medium variance processes. These are exactly the processes we seek, so the error of the reconstructions can be used to find them. In particular, changes in the structure of the error over time serves as an indicator of their presence.

To better understand why reconstruction error can be used to find features, consider Fig 2, which reconstructions for several choices of $D$ during the breakdown of the leading wave in the dual pycnocline data set. As $D$ increases it is clear that large variance processes are approximated first, followed by smaller and smaller processes. As expected the EOF reconstruction effects multiple processes simultaneously. A choice of $D$ near 1 corresponds to capturing processes with large variance such as the wave guide. Intermediate choices for $D$ capture the large variance structures and some, but not all of the medium variance structures. Short to medium duration processes of interest such as the breakdown of the leading wave are poorly

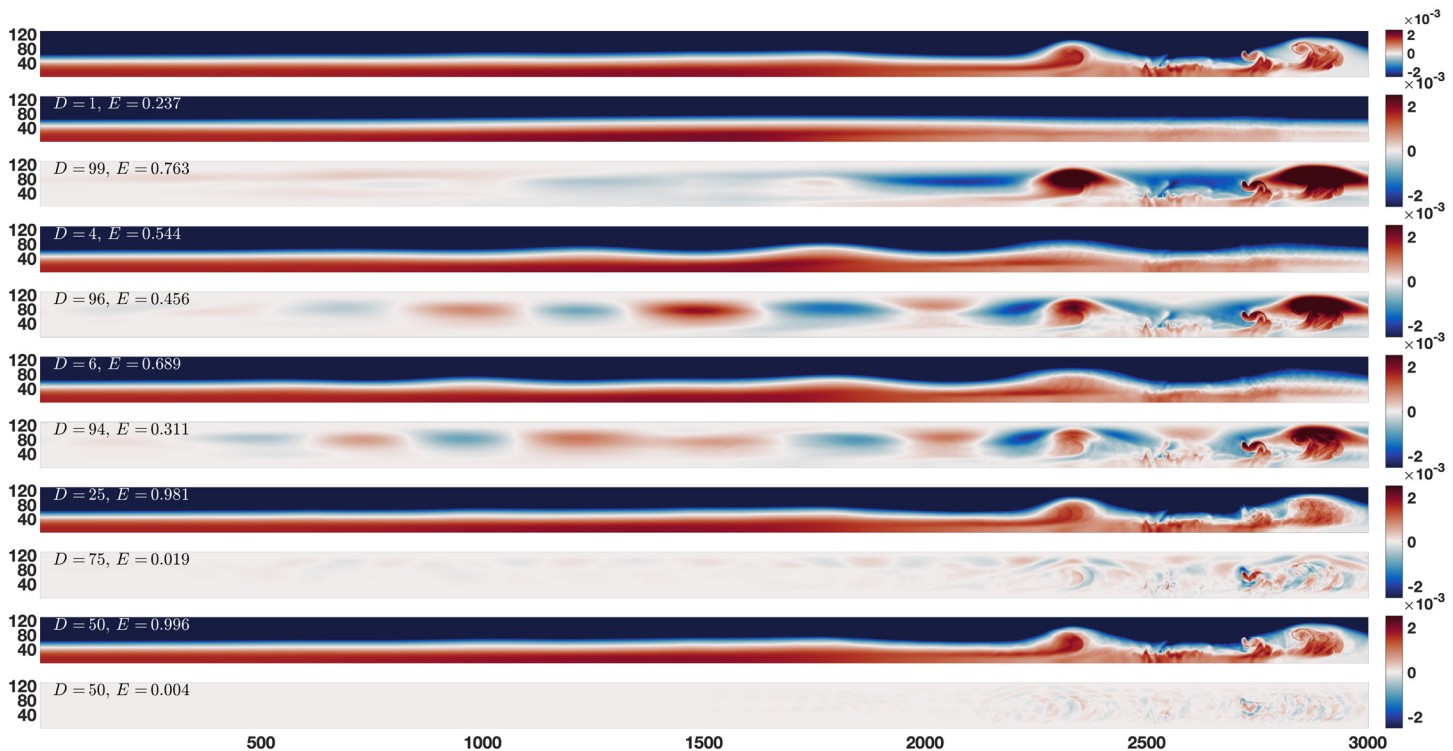

**Fig 2. Error of reconstructions.** Continually increasing choices of $D$ at time output 80 in the density field (first 3 choices are the obvious elbow test choices). This time was chosen to look at the breakdown of the wave, which is a medium variance event with a variety of scales of structures. The top panel is the data, while reconstruction and reconstruction error are in pairs below it for comparison, with $D = 1, 4, 6, 25, 50$ increasing downward. As $D$ increases the wave guide is approximated first, followed by lower variance structures like the breakdown, and finally the fine details of of the breakdown. By $D = 25$ the large variance wave guide is well approximated, but more modes are required to capture the fine details of the breakdown. By $D = 85$ (not shown) there is almost no error anywhere.

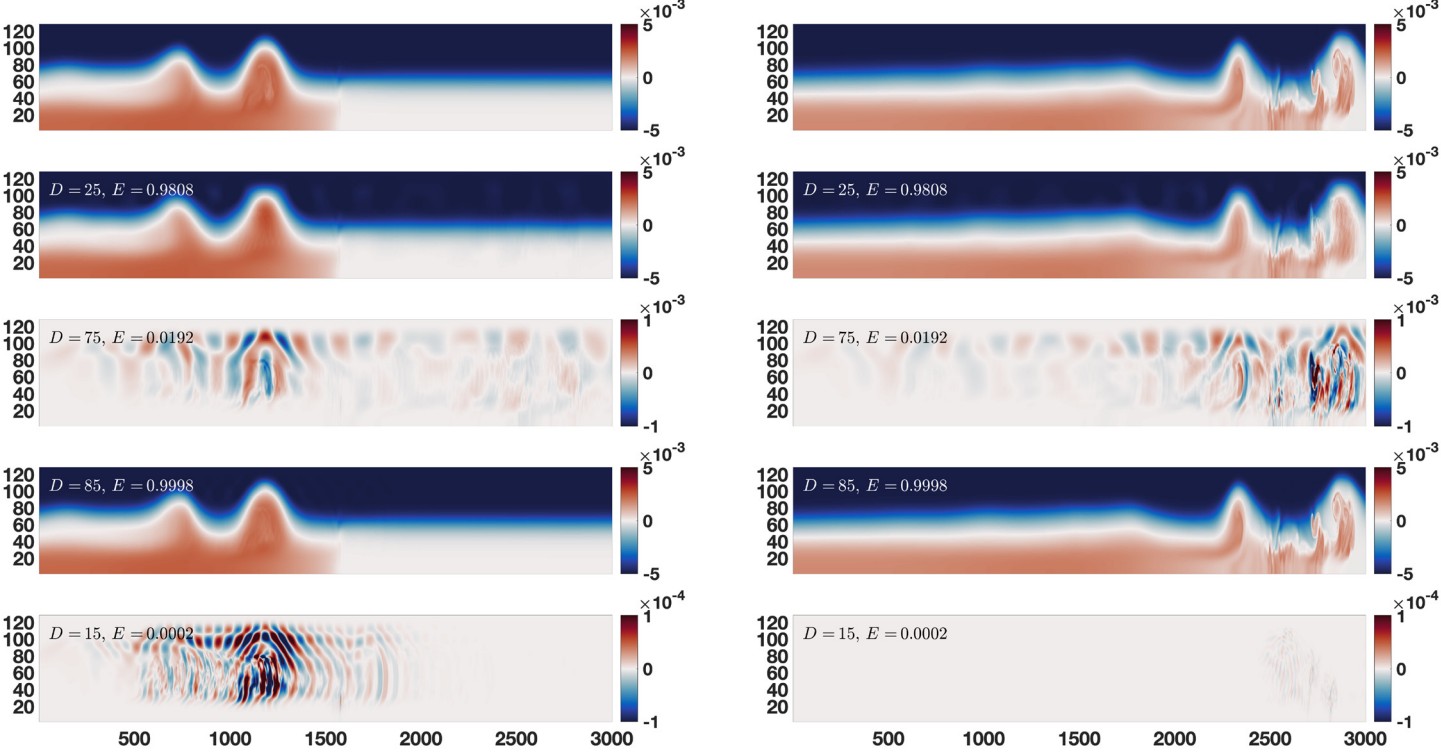

**Fig 3. Error of reconstructions over time.** Two examples of changes in error of reconstructions over time: the left panel is at time 20 and the right is at time 80. Similar to Fig 2, top panels are the data, while in pairs underneath we have $D$ = 25, 85 reconstructions and reconstruction errors. For a 25 mode reconstruction the infinity norm error is greater during the shedding (right) than at time 20 (left). This is because for this lower number of modes, the medium variance shedding event has not yet been fully captured. For an 85 mode reconstruction the reverse is true: the error is higher during the early time. This is because for this higher number of modes, the medium variance event has been almost fully captured, and now the very small variance structures in the early times are left (note the change in error scale between the 25 mode and 85 mode reconstructions).

approximated for some intermediate $D$ values, but as $D$ increases the breakdown is also well approximated. Finally, a choice of $D$ near $N$ corresponds to an approximation which misses only noise.

While Fig 2 shows multiple choices for $D$ at a single time, Fig 3 gives an example for two different times and two different choices for $D$, in order to give some sense of the change in error over time for a fixed $D$ value. We see that for a low number of modes the error increases during the medium variance breakdown event. This is because the larger variance background state and propagation processes have taken precedence in the reconstruction. We also see that for a high number of modes the error goes down at the time of the breakdown event. This is because there are so many modes in the reconstruction that medium variance events like the breakdown have been well approximated, and the processes that are left are virtually noise.

Together, Figs 2 and 3 show that the medium variance processes of interest are poorly approximated for some intermediate values of $D$. Since these are the processes we are interested in, we can look at the error of the reconstructions to identify when they occur. When error is high for a short time, it can indicate the presence of dynamics worthy of further study. Rather than attempt to determine a single intermediate choice for $D$ which will help identify times of interest, we simply calculate the error of the reconstruction for every choice of $D$, and for all times. In order to collapse the error information to a more manageable and interpretable size, we use a norm of the time slice error, rather than a full error plot like those in Figs 2 and 3. Moreover if we use the L2 norm at every time slice the error's distribution is unknown, and

may be spread thin over the whole domain or concentrated in some way. To avoid this ambiguity we use the infinity norm to make interpretation more straightforward. The error map $\epsilon_D(t_j)$ of an EOF reconstruction with $D$ modes at time $t_j$ is given by

$$
\begin{aligned}
\epsilon_D(t_j) &= \left| \mathbf{x}(t_j) - \left( \sum_{k=1}^{D} a_k(t_j)\phi_{\mathbf{k}} + \langle \mathbf{x} \rangle \right) \right|_{\infty} \\
&= \left| \sum_{k=1}^{\min\{M,N\}} a_k(t_j)\phi_{\mathbf{k}} + \langle \mathbf{x} \rangle - \left( \sum_{k=1}^{D} a_k(t_j)\phi_{\mathbf{k}} + \langle \mathbf{x} \rangle \right) \right|_{\infty} \\
&= \left| \sum_{k=1}^{\min\{M,N\}} a_k(t_j)\phi_{\mathbf{k}} - \sum_{k=1}^{D} a_k(t_j)\phi_{\mathbf{k}} \right|_{\infty} \\
&= \left| \sum_{k=D+1}^{\min\{M,N\}} a_k(t_j)\phi_{\mathbf{k}} \right|_{\infty}
\end{aligned}
\tag{10}
$$

for each $t_j$. This is simply the infinity norm of the modes excluded from a reconstruction with $D$ modes at every time step. By construction $\epsilon_D(t_j)$ is a function of both time and the number of modes used in the reconstruction $D$. We call this function the error map for the EOF reconstructions of the data set, or simply "the error map." The number of modes produced by an EOF analysis is min{M,N}. The error map is therefore of size min{M,N} × N. In the case of CFD data sets $M > N$, and so the error map has size $N \times N$. In practice, forming the error map is computationally inexpensive, as $N$ tends to be small. The computations are simply an SVD decomposition, and one norm calculation for every time output and for every choice of $D$. In many contexts it is standard practice to perform an EOF analysis anyway, in which case the EOF error map is easily derived from the existing reconstructions.

## 3 Results

Although the method developed in this manuscript may be applied to any time-indexed model output for which an EOF analysis would be appropriate, we will consider concrete examples from three qualitatively different simulations in stratified flow dynamics. It is not necessary that the reader have training in fluid dynamics to understand the method presented, but we provide background for each of the data sets for those who are interested. In order to keep a consistent focus, and because the varying density is the essential component of stratified flows, we will focus on the dynamics of density. As discussed in the introduction, in practice the error map method would be used to identify features in all variables within the data set. For expository purposes, we have elected to present our method on one variable in multiple flows, rather than on multiple variables in one flow.

All three data sets are simulated using a spectral collocation method (SPINS [28]). Grid doubling/halving experiments were performed to ensure that the numerical results were robust. The details of the physics of the dual pycnocline and collision cases will be discussed in future publications, while the details of the spontaneous instability case may be found in [29]. As the focus here is on the data analysis method presented, all data sets throughout this manuscript are presented in terms of grid points, time output number, and numeric field values. All MATLAB codes for production of these figures, along with all data sets, are included in the supplementary information.

For reference, the normalized scree of the first thirty modes for all three data sets are plotted in Fig 4. Note that these three scree are plotted together, but that the total number of modes

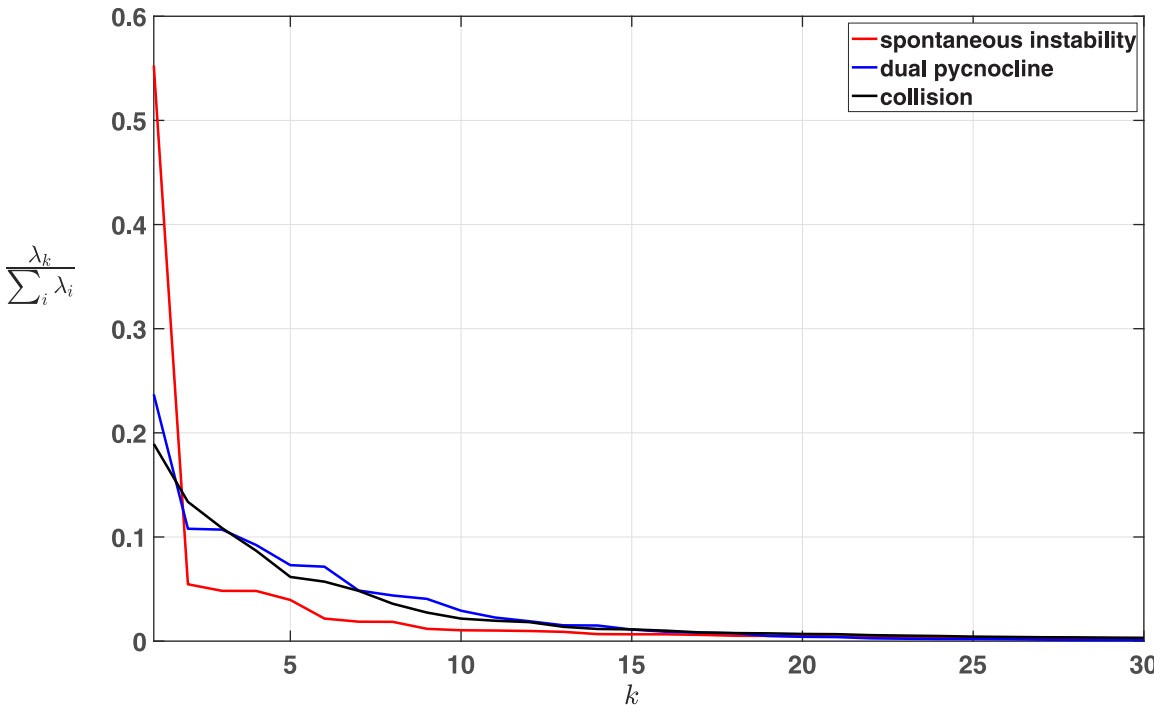

**Fig 4. Scree plots.** Each scree is a plot of the normalized eigenvalues as a function of mode $k = 1, \ldots, 30$, the $k$ being the mode index from Eq 7. The sum in the normalization is over all eigenvalues of the given dataset. See text for details.

differs by case. The spontaneous instability data set has 131 total modes, the dual pycnocline data set has 100, and the collision data set has has 150. The fast convergence of the eigenvalues is clear in each case. Clearly the spontaneous instability has the most variance in the first few modes, while the dual pycnocline and collision cases have more variance in higher modes.

We now discuss the error maps $\epsilon_D(t_j)$ for each of the data sets under consideration.

## 3.1 Spontaneous instability

The first data set is the spontaneous shear instability of a very large amplitude internal solitary wave, studied in detail in [29], following previous related work [30], [31]. Here the flow is initialized from a solution to the Dubreil–Jacotin–Long (DJL) equation, which is formally equivalent to the stratified Euler equations. The initial wave develops a spontaneous instability at the rear of the wave. The instability grows and eventually exits the wave. Detailed discussion, including the effects of three-dimensionalization can be found in [29]. See the top four panels of Fig 5 for a visual representation of the density field's evolution in this case. The internal solitary wave serves as a "base" flow with the spontaneous shear instability playing the part of a temporary perturbation. This data set is thus close to classical hydrodynamic instability theory, for which a base flow and a perturbation are specified analytically, but still requires a full integration of the stratified Navier-Stokes equations for a full description since a purely analytical treatment is not possible in this case. In what follows this case will be referred to as the "spontaneous instability" case.

The bottom panel of Fig 5 shows the results of applying the error map method to the spontaneous instability data set. For times less than $t = 50$ there is very little error due to the stable background profile's large variance. This means even a reconstruction with $D = 1$ has small error over this time period. This is consistent with the large first eigenvalue (Fig 4). As the

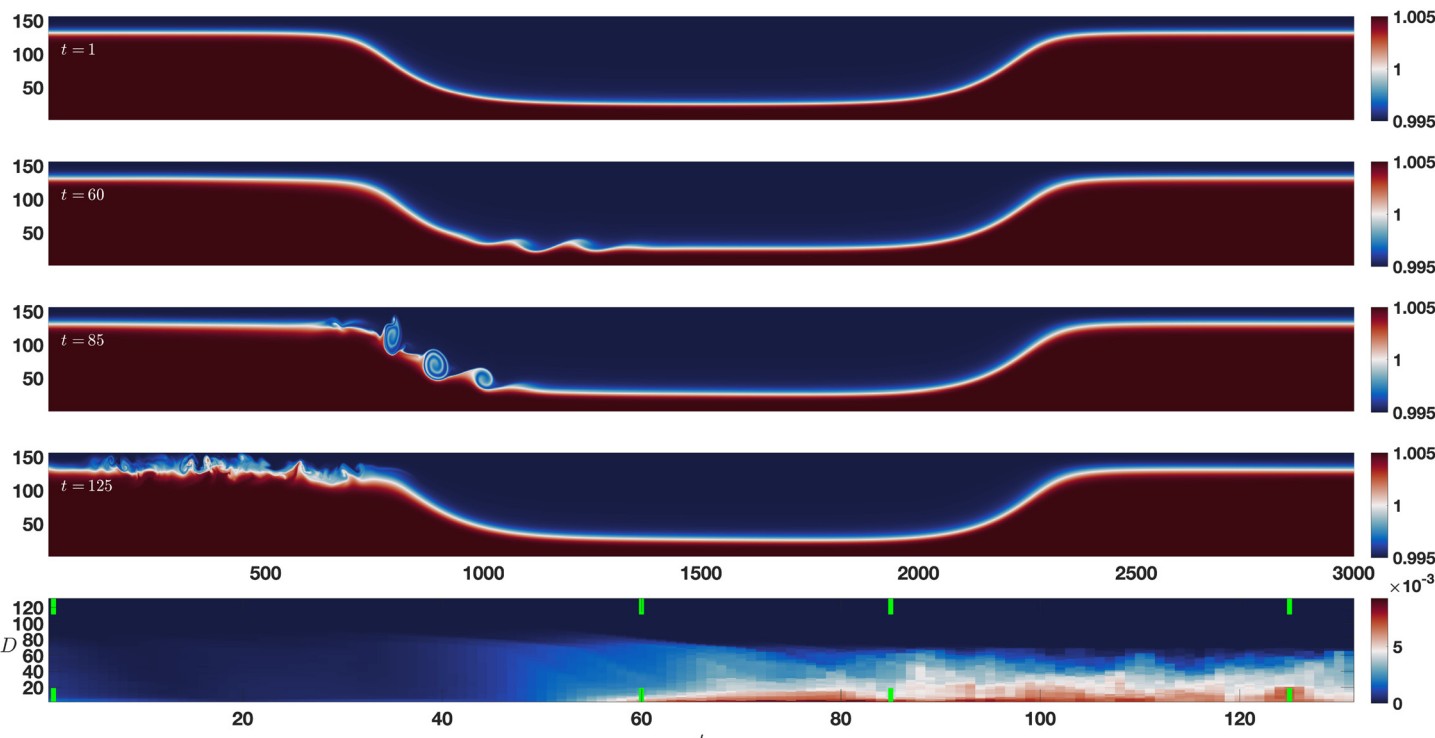

**Fig 5. Spontaneous instability error map.** A spontaneous shear instability forms and evolves, with time increasing from the top to the bottom of the first four panels. The bottom panel is the error map with time increasing left to right, and vertical axis of increasing $D$, with pairs of vertical green lines indicating the times of the upper panels as time increases from left to right. See text for details.

instability develops, we see error in the reconstructions for small to intermediate values of $D$. This is due to the instability's low variance (and therefore priority) relative to the background profile, as discussed in section 2.3. This error continues to the end of the simulation as the instability evolves. The error map clearly indicates the presence of the instability as a time period of interest in the data set, as indicated in the obvious change in the structure of the error over time.

### 3.2 Dual pycnocline

The second data set we examine is a simulation of an internal wave train in a spatially varying wave guide, generated by what experimentalists refer to as a lock release: fluid of a set density is suddenly released from behind a barrier and is allowed to freely form waves in the stratified tank. The particular situation is set up so that a wave train of internal solitary waves with a trapped core forms, propagates some distance and then encounters a sharp change in the background density (a pycnocline). This change removes the near bottom stratification, while the main wave guide remains unchanged. To the best of our knowledge, there is no *a priori* theory for the wave evolution in this cases and we find that the change in the near bottom wave guide leads to the destruction of the trapped core in the leading wave. This in turn leads to a significant increase in short length scale activity and a loss of material from the leading wave, and a significant perturbation to the second wave in the wave train. Unlike the spontaneous instability data set, in this case there is no readily apparent way to define a "base" flow in this case since even prior to the collapse of the core, the disappearance of the near boundary wave guide implies a core cannot persist [32]. See the top four panels of Fig 6 for a visual representation of

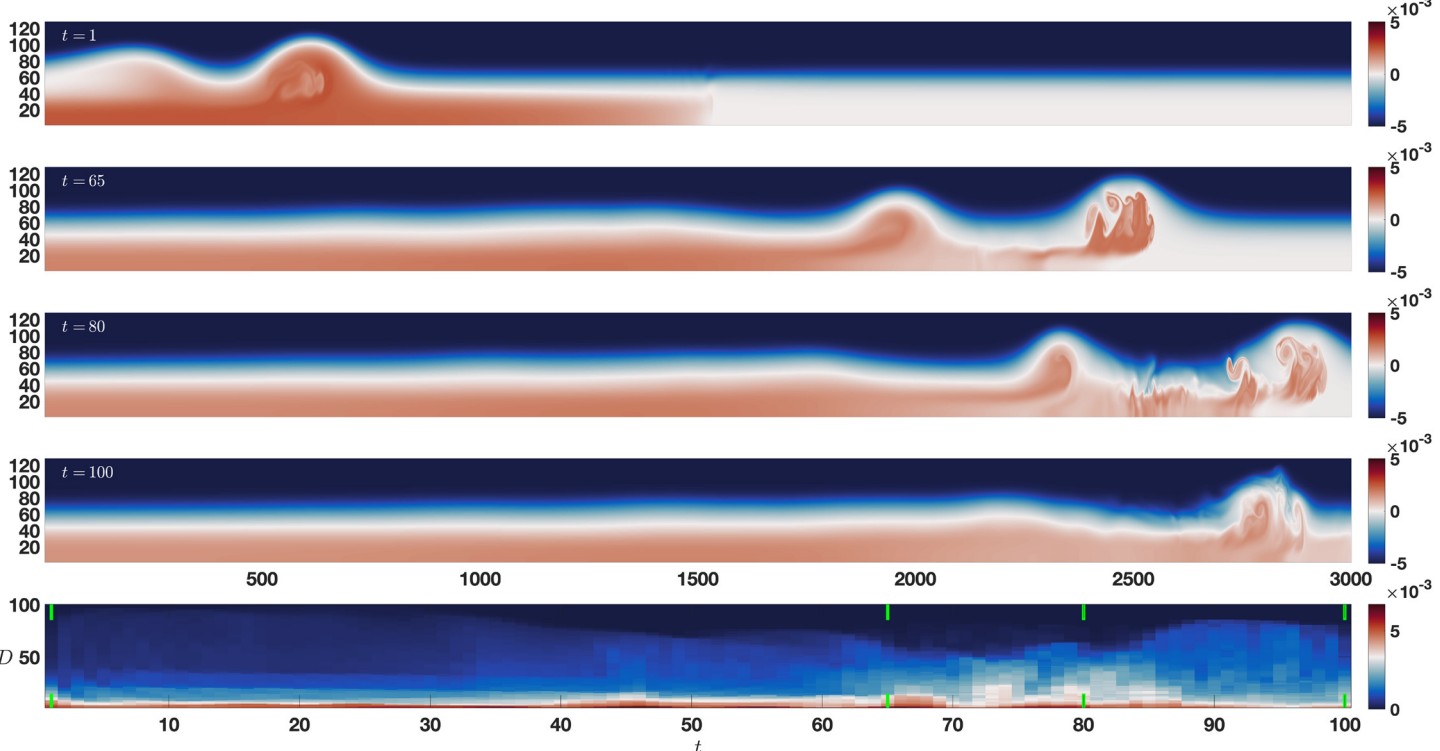

**Fig 6. Dual pycnocline error map.** An internal wave train propagates from left to right and encounters a sharp change in the background density profile. The bottom panel is the error map with time increasing left to right, and vertical axis of increasing $D$, with pairs of vertical green lines indicating the times of the upper panels as time increases from left to right. See text for details.

the density field's evolution in this case. The dynamics are considerably more complex than the spontaneous instability dataset, and there is no obvious tie in with classical stability theory. This case thus acts as a stress test for our analysis method. In what follows this case will be referred to as the "dual pycnocline" case.

The bottom panel of Fig 6 shows the results of applying the error map method to the dual pycnocline data set. The clearest error structure is during the shedding event of the leading wave beginning around $t = 65$, up until the leading wave leaves the domain around $t = 90$. The change in structure of the error map with increasing $D$ during this time period corresponds to the rank ordering of processes by variance illustrated in Fig 2 at $t = 80$. Once again the error map clearly indicates a time period of interest through the changes in the structure of the error over time.

The observant reader may have noticed the persistent error for low values of $D$ in Fig 6 which was not present in Fig 5. The EOF modes are functions of space but not time, so propagating structures require multiple modes. This is analogous to the way a sequence of hand drawn stills can be used to create an animation, despite each picture being a functions of space only. The propagation of the basic internal waves/gravity current structure is an example of a medium scale process that lasts the duration of the simulation, requiring a minimum amount of modes to even roughly approximate. This is consistent with the scree in Fig 4, which shows that more variance is found in higher modes than in the spontaneous instability case. As a result there is persistent error for low choices of $D$ even before the wave train encounters the density change around $t = 35$. This is in sharp contrast to the spontaneous instability case there was almost no propagation of the steady background state, and so even a one mode

reconstruction had low error. Similarly, the slight increase in error from $t = 40$ to $t = 65$ is due to the instability in the lead wave induced by interaction with the density change. There are more small scale processes present during this time, requiring more modes to approximate those processes well.

### 3.3 Collision

The third data set we examine involves the repeated collision of mode-1 (i.e. all lines of constant density rise and fall together) and mode-2 (i.e. lines of constant density above a given height rise, while those below fall, forming a lump-like wave) internal solitary waves in a two pycnocline stratification. This simulation is constructed based on the observations in [33] that suggest mode-mode collisions can irreversibly deform the higher mode. By choosing a double pycnocline we ensure that the interaction takes place without significant instability and three-dimensionalization. This allows us to confirm that our analysis method is capable of capturing nonlinear phenomena loosely linked to the concept of solitons, as opposed to turbulent transition. See the top four panels of Fig 7 for a visual representation of the density field's evolution in this case. The dynamics are complex, but compared to the spontaneous instability and dual pycnocline cases, there are no instances of short scale instabilities, and no turbulence develops. In fact, the complex pattern of constructive and destructive interference between the waves would make an analysis method based on kinetic energy or vorticity very difficult to interpret. This case thus acts as a different test for our analysis method, since the nonlinear effects in this

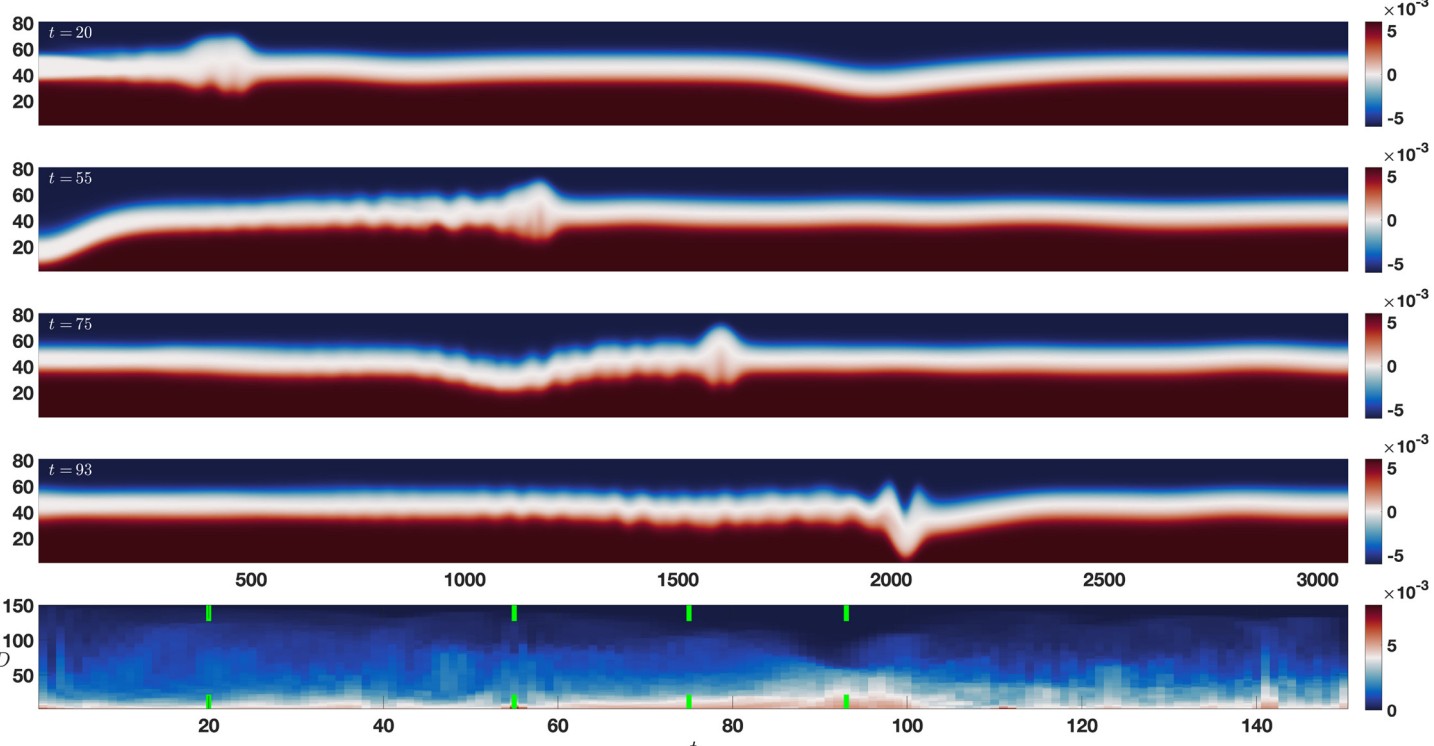

**Fig 7. Collision error map.** The repeated collision of a mode-1 wave with a mode-2 wave. Initially (top panel), the mode-2 wave propagates slowly from left to right, and the mode-1 wave propagates quickly from right to left. At $t = 55$ the mode-1 reflects from the left wall, as the mode-2 continues propagation to the right. At $t = 75$ the mode-1 wave has almost overtaken the mode-2 wave as both propagate to the right. At $t = 93$ the two waves nearly coincide. The bottom panel is the error map with time increasing left to right, and vertical axis of increasing $D$, with pairs of vertical green lines indicating the times of the upper panels as time increases from left to right. See text for details.

case involve soliton–like behaviour that becomes evident during collisions (both wave–wave and wave–wall). In what follows this case will be referred to as the "collision" case.

The bottom panel of Fig 7 shows the results of applying the error map method to the collision data set. The waves are initialized so that the mode-2 wave is travelling rightward and the mode-1 wave is travelling leftward. As discussed for the dual pycnocline case, multiple modes are required for propagation, but in this case there is propagation of two different waves at two different speeds. This double propagation requires many modes, and again Fig 4 shows the variance in higher modes. The smaller error anomalies correspond to reflections from the boundary: the mode-1 wave at $t = 55$ and $t = 111$, and the mode-2 wave at $t = 141$. The large error anomaly from $t = 60$ to $t = 100$ corresponds to the overtaking of the mode one wave by the mode two wave. The clear error structure around $t = 90$ to $t = 95$ corresponds to the superposition of the two waves. As in the other two cases, we again see that the error map clearly indicates features in the data set.

## 4 Discussion

The EOF error map identified time periods of interest in each of the three cases presented in section 3. The method was successful even though only one of the three data sets had a classical "background–perturbation" split. And while the collision data set featured a complex patterns of constructive and destructive interference, making the kinetic energy and vorticity evolution very difficult to interpret, the error map method was still successful. Note that these two dimensional data sets were chosen so that the error map could be easily visualized alongside time outputs for expository purposes. The error map method still identifies features even if the data set is so large that it is otherwise difficult to visualize. Moreover, because the error map method collapses all non-time dimensions for a given reconstruction and time output, the method can be applied to any time-indexed model output, provided an EOF decomposition is appropriate and computationally feasible.

For very large data sets, there are alternatives to reduce the computational burden. In particular it is clear that in many cases the full error map is unnecessary. For completeness we included reconstruction of all possible $D$ values in the Figures of section 3. However the error structures would have been clear with fewer modes than the maximum. In particular, for half as many modes as the maximum we could have drawn all of the same conclusions. This is unsurprising given the convergence of the eigenvalues in all cases (Fig 4). Of course, given the steady increase in computational power, some data sets will be too large to fit into memory. However even here, a rapidly developing literature offers a way to compute the error map, albeit with an added burden of increased computational time [34], [35].

In the examples given here, error maps were calculated only for model output of consistent physical units. Our code [28] outputs multiple physical fields, and we chose to focus on only the density fields. As a result the EOF was carried out on a physical field with only one type of physical unit. Care must be taken if the model output includes data with different units. While multiple data types may be included together in an EOF reconstruction, the non-uniform units cause differing weights of importance on the different data types. Scalings may be chosen to attempt to correct this, but the more types of units in a data set, the more relative scalings must be considered. Moreover these scalings can have a profound effect on the resulting EOF reconstructions. All of this is a general principle when carrying out an EOF analysis. In particular, for the error map method, the relative scalings effect the reconstructions, and therefore the error maps as well. This scaling problem is most easily solved by avoiding it altogether: simply carry out a separate EOF analysis on each data type in the model output, as was done here.

The error map method has several possible extensions. For example, reconstructions from using one of the many modifications of EOF (see [5]) could be employed or a different norm chosen to measure the error. Although the focus here was on time-indexed model output, a spatial dimension could also be used as the index. In that case the method identifies spatial extents of interest, and the error map would be a function of the spatial dimension and $D$. In general, any dimension of a data set may be used as an index for the method, provided continuous subsets of that dimension have a useful interpretation. Such extensions are possibilities for future work.

The error map method also serves as a replacement for rough heuristics such as "the elbow test" [8] for deciding how many modes to keep. Modes with low energy, which may easily be removed by a standard elbow test, may still represent important dynamics [24]. In particular unstable modes start small but grow to be very important to the dynamics. In order to avoid missing dynamically relevant modes, simply pick a value of $D$ large enough to avoid significant error structures in the map. This corresponds to picking the lowest row in the error map which has no significant error at any time.

## 5 Conclusion

EOF error maps identify time periods of interest in time-indexed model output which are worthy of further study. The method is easily implemented and computationally inexpensive. EOF error maps are appropriate for any data set for which an EOF analysis is appropriate. In the case of CFD data sets, this typically means many domain snapshots of a single physical field. We also recently published the $\gamma$ method [36], which was designed primarily to find features in data sets consisting of time series sampling multiple physical fields. Together these two methods allow the quick identification of interesting features in a wide variety of data sets.

## Supporting information

**S1 File. MATLAB code for Fig 4.**
(M)

## Acknowledgments

Thanks to Andrew Grace for assistance in editing.

## Author Contributions

**Conceptualization:** Justin Shaw, Marek Stastna.

**Data curation:** Justin Shaw, Marek Stastna.

**Funding acquisition:** Justin Shaw, Marek Stastna.

**Investigation:** Justin Shaw.

**Methodology:** Justin Shaw.

**Project administration:** Marek Stastna.

**Resources:** Marek Stastna.

**Supervision:** Marek Stastna.

**Validation:** Justin Shaw, Marek Stastna.

**Visualization:** Justin Shaw.

**Writing – original draft:** Justin Shaw.

**Writing – review & editing:** Marek Stastna.

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
