## [Decision Letter · Decision Letter 0]

10 Sep 2019

PONE-D-19-21139

Feature identification in time-indexed model output

PLOS ONE

Dear Mr. Shaw,

Thank you for submitting your manuscript to PLOS ONE. After careful consideration, we feel that it has merit but does not fully meet PLOS ONE’s publication criteria as it currently stands. Therefore, we invite you to submit a revised version of the manuscript that addresses the points raised during the review process.

Please address reviewer comments. In particular, focus on improved discussion of the methods, interpretability, and performance of the techniques to show improvement over the comparison discussions. Additionally, great detail should be given to the manuscript formatting, grammar, and ordering.

We would appreciate receiving your revised manuscript by Oct 25 2019 11:59PM. To enhance the reproducibility of your results, we recommend that if applicable you deposit your laboratory protocols in protocols.io, where a protocol can be assigned its own identifier (DOI) such that it can be cited independently in the future. For instructions see: http://journals.plos.org/plosone/s/submission-guidelines#loc-laboratory-protocols

We look forward to receiving your revised manuscript.

Kind regards,

Bobak Mortazavi

Academic Editor

PLOS ONE

Journal Requirements:

'This work was funded by a Natural Sciences and Engineering Research Council of 358

Canada grant (RGPIN-311844-37157) and a PGS-D. Thanks to Andrew Grace for 359

assistance in editing.'

'The author(s) received no specific funding for this work.'

Additional Editor Comments (if provided):

Reviewers' comments:

Reviewer's Responses to Questions

**Comments to the Author**

1. Is the manuscript technically sound, and do the data support the conclusions?

Reviewer #1: Yes

Reviewer #2: Partly

Reviewer #3: Yes

2. Has the statistical analysis been performed appropriately and rigorously? 

Reviewer #1: Yes

Reviewer #2: No

Reviewer #3: Yes

3. Have the authors made all data underlying the findings in their manuscript fully available?

Reviewer #1: Yes

Reviewer #2: Yes

Reviewer #3: Yes

4. Is the manuscript presented in an intelligible fashion and written in standard English?

Reviewer #1: Yes

Reviewer #2: Yes

Reviewer #3: No

5. Review Comments to the Author

Reviewer #1: This paper describes a method to identify time points of interest in time series data via the reconstruction error of EOF. The method is sound and sufficient details are provided. The manuscript can be improved by the following points:

- The numbering of the sections throughout the manuscript is off. Please check to make sure they reflect the actual numbering. e.g. last paragraph of section 1, all numbers are off.

- The structure of the manuscript can be reordered. The introduction of the datasets prior to the methods can be distracting since the focus of the manuscript is on the method. Also, the content of EOF can be reduced as this is not the innovation proposed by the authors. The authors should mention clearly what the contributions are compared to the existing literature.

- The results of this proposed method should be compared with some baseline methods in the literature. They should be applied to the same datasets and metrics of performance evaluation and comparison should included.

Reviewer #2: This paper proposes a method for identifying time periods of interest in time-series data by using empirical orthogonal functions (EOF) or principal component analysis. Leveraging PCA to determine important modes in the specific application of fluid dynamics may be new but this reviewer is concerned with the novelty in the proposed method. PCA have been extensively used to determine important components of different signals. However, the current paper does not provide any comparison with the state-of-the-art algorithms in this area. Moreover, there are a few parameters such as D in this method that needs configuration for which the authors are not providing a systematic solution; therefore, it is not clear in the paper how others can use this method to find important modes/components within a time-series signal. These two concerns makes the paper inappropriate to being published in the current form; Below are the detailed comments that need to be addressed:

1- The authors need to provide comparison with the state-of-the-art.

2- Setting of parameter D should be clearly explained.

3- The authors indicated that the proposed method "can minimize the cost of uptake and maximize the clarity of the presentation"; however, there is no results to support these claims. Please provide additional results/discussion about this.

4- The authors provided a great deal of discussion in the introduction section about inability of the current methods regarding providing interpretable representations that carry physical meanings. However, there is no sign of interpretability in the proposed method. It seems that the proposed method does not explain physical phenomena either.

5- Most of the figures do not have axis labels and physical units.

6- Most of the formulation provided in section 3 are the typical equations used in PCA/EOF; the authors need to clarify which parts are novel, where are the gaps that they are addressing, and where and how they have improved the current formulation.

7- In the text figure 7 has referred before figure 5. Fix the ordering.

8- For the results section show how the "time period of interest" is detected with the proposed method.

9- Provide a few examples about the application of determining "time period of interest", especially, in applications other than fluid dynamics.

Reviewer #3: In this study, the authors gave an analysis on Feature identification in time-indexed model output. The introduction provides a good, generalized background of the topic. The following points should be clarified:

1- The motivations for this study need to be made clearer.

2- However, to make the motivation clearer and to differentiate the paper some more from other applied papers, the author may wish to provide examples of some of the applications.

3-The authors must be explained how their method is novel and on what basis (justifications).

4- The authors did not mention the importance of Feature identification in time-indexed.

5- Also the authors are advised to update the manuscript before final submission as it contains some typos and grammatical errors.

6- The authors claim that their method is easily implemented and computationally inexpensive. It should be explained in detail for the interest of reader make it more interesting.

My recommendation is a Major revision.

6. PLOS authors have the option to publish the peer review history of their article (what does this mean?). If published, this will include your full peer review and any attached files.

Reviewer #1: No

Reviewer #2: No

Reviewer #3: No

---

## [Author Response · Author response to Decision Letter 0]

10 Oct 2019

Please see the accompanying response to reviewers pdf.

---

## [Decision Letter · Decision Letter 1]

6 Nov 2019

Feature identification in time-indexed model output

PONE-D-19-21139R1

Dear Dr. Shaw,

We are pleased to inform you that your manuscript has been judged scientifically suitable for publication and will be formally accepted for publication once it complies with all outstanding technical requirements.

With kind regards,

Bobak Mortazavi

Academic Editor

PLOS ONE

Additional Editor Comments (optional):

Reviewers' comments:

Reviewer's Responses to Questions

**Comments to the Author**

1. If the authors have adequately addressed your comments raised in a previous round of review and you feel that this manuscript is now acceptable for publication, you may indicate that here to bypass the “Comments to the Author” section, enter your conflict of interest statement in the “Confidential to Editor” section, and submit your "Accept" recommendation.

Reviewer #1: All comments have been addressed

Reviewer #2: All comments have been addressed

Reviewer #3: All comments have been addressed

2. Is the manuscript technically sound, and do the data support the conclusions?

Reviewer #1: (No Response)

Reviewer #2: Yes

Reviewer #3: Yes

3. Has the statistical analysis been performed appropriately and rigorously? 

Reviewer #1: (No Response)

Reviewer #2: Yes

Reviewer #3: Yes

4. Have the authors made all data underlying the findings in their manuscript fully available?

Reviewer #1: (No Response)

Reviewer #2: Yes

Reviewer #3: Yes

5. Is the manuscript presented in an intelligible fashion and written in standard English?

Reviewer #1: (No Response)

Reviewer #2: Yes

Reviewer #3: Yes

6. Review Comments to the Author

Reviewer #1: (No Response)

Reviewer #2: Thanks for fully addressing all the comments. The only minor point is to add the axis titles to all the figures. That is true that the authors described this in the paper, but the figures should always stand alone and the reader should not read all the paper to understand them.

Reviewer #3: The authors studied on the Feature identification in time-indexed model output. All comments have been addressed so paper is accepted.

7. PLOS authors have the option to publish the peer review history of their article (what does this mean?). If published, this will include your full peer review and any attached files.

Reviewer #1: No

Reviewer #2: No

Reviewer #3: No

---

## [Editor Report · Acceptance letter]

18 Nov 2019

PONE-D-19-21139R1 

Feature identification in time-indexed model output 

Dear Dr. Shaw:

I am pleased to inform you that your manuscript has been deemed suitable for publication in PLOS ONE. Congratulations! Your manuscript is now with our production department. 

With kind regards,

on behalf of

Dr. Bobak Mortazavi 

Academic Editor

PLOS ONE